# Advances in Rubber Compounds Using ZnO and MgO as Co-Cure Activators

**DOI:** 10.3390/polym14235289

**Published:** 2022-12-03

**Authors:** Md Najib Alam, Vineet Kumar, Sang-Shin Park

**Affiliations:** School of Mechanical Engineering, Yeungnam University, 280, Daehak-ro, Gyeongsan 38541, Republic of Korea

**Keywords:** rubber, cure activators, cross-linking, synergism, tensile properties, thermal properties

## Abstract

Zinc oxide performs as the best cure activator in sulfur-based vulcanization of rubber, but it is regarded as a highly toxic material for aquatic organisms. Hence, the toxic cure activator should be replaced by a non-toxic one. Still, there is no suitable alternative industrially. However, binary activators combining ZnO and another metal oxide such as MgO can largely reduce the level of ZnO with some improved benefits in the vulcanization of rubber as investigated in this research. Curing, mechanical, and thermal characteristics were investigated to find out the suitability of MgO in the vulcanization of rubber. Curing studies reveal that significant reductions in the optimum curing times are found by using MgO as a co-cure activator. Especially, the rate of vulcanization with conventional 5 phr (per hundred grams) ZnO can be enhanced by more than double, going from 0.3 Nm/min to 0.85 Nm/min by the use of a 3:2 ratio of MgO to ZnO cure activator system that should have high industrial importance. Mechanical and thermal properties investigations suggest that MgO as a co-cure activator used at 60% can provide 7.5% higher M100 (modulus at 100% strain) (0.58 MPa from 0.54 MPa), 20% higher tensile strength (23.7 MPa from 19.5 MPa), 15% higher elongation at break (1455% from 1270%), 68% higher fracture toughness (126 MJ/m^3^ from 75 MJ/m^3^), and comparable thermal stability than conventionally using 100 % ZnO. Especially, MgO as a co-cure activator could be very useful for improving the fracture toughness in rubber compounds compared to ZnO as a single-site curing activator. The significant improvements in the curing and mechanical properties suggest that MgO and ZnO undergo chemical interactions during vulcanization. Such rubber compounds can be useful in advanced tough and stretchable applications.

## 1. Introduction

The discovery of rubber mastication in the year 1821 by Hancock and the rubber vulcanization by Charles Goodyear in the year 1839 revolutionized the industrial utility of rubbers. Without vulcanization, rubbers remain stiff in cold weather and sticky in hot weather, which restricts their industrial applications. Charles Goodyear first started the vulcanization of rubber by simply heating rubber with sulfur. With this process, vulcanization takes a longer time and is currently uneconomic for industrial applications. In modern vulcanization systems, many ingredients have been used for the vulcanization of rubber. Among them, sulfur, accelerator, and activator are the basic ingredients. With these vulcanizing ingredients, cure activators play an important role in regenerating precursors that can effectively cross-link the rubber. A combination of metallic oxide and fatty acid acts as a cure activator. Currently, the combination of zinc oxide and stearic acid is the most successful cure activator system in the rubber industries. Generally, zinc oxide at 2 to 5 phr (per hundred grams of rubber) and stearic acid at 0.5 to 3 phr are used as cure activators in the vulcanization of rubber [1,2]. Mostly, 5 phr zinc oxide along with 2 phr stearic acid is the conventional amount in the tire industries to achieve a better modulus, low heat build-up, and good abrasion resistance properties [3]. Zinc oxide in rubber compound also acts as an adhesion promoter between the interfaces of brass-coated steel cords and the rubber in radial tires [4]. About 10^5^ tons of zinc oxide are produced annually, of which 50–60% is used in rubber industries [5]. Zinc oxide is well known as a high carcinogen for aquatic organisms and thus an environmental pollutant. Importantly, soluble zinc compounds are toxic to aquatic species [6]. The amount of zinc oxide can be reduced by using higher surface-active zinc oxide such as nano zinc oxide. However, recent toxicological studies suggest that even nano zinc oxide is more toxic directly or by dissolution than zinc ion or some combination thereof [7,8]. When rubber products are thrown into the environment after the end of their use, zinc oxide releases into the environment during degradation or by leaching from the landfill sites. Zinc oxide release to the environment by leaching should enhance the ecosystem exposure, even though it is difficult to measure. Modeling efforts suggest that the zinc oxide content presently is high in wastewater treatment plant effluent and can cause toxicological risk to aquatic species [9]. Another major source of zinc oxide in the environment was detected from tire wear during the service life [10]. Hence, to relieve the environmental pollution, either an alternative cure activator that is less toxic or at least the amount of zinc oxide should be reduced in the rubber formulation.

To solve this environmental issue, the reduction of zinc oxide amount was first considered through zinc-based materials [5,11,12,13,14,15,16], because zinc ion is almost necessary for the formation of a zinc-accelerator complex that can effectively cross-link the rubber chains. For example, layered double hydroxides or zinc-containing clays can effectively reduce the level of zinc oxide [5,11,12,13], but they are less dispersible in the pristine form to the non-polar rubber and result in reduced physical properties which are economically unfavorable [17]. Some researchers found some zinc complexes [18,19,20,21] might be the alternative, but in some selective rubbers. Nono zinc oxide or nano zinc hydroxide [22,23,24,25,26,27,28,29,30] could be the alternative but are relatively expensive. Wu et al. investigated carbon nanodots as an alternative eco-friendly cure activator for sulfur-based rubber vulcanization and found some promising results in diene rubbers [31]. Other metal oxides such as CaO, MgO, CdO, CuO, PbO, and NiO can be used as cure activators. Among different metal oxides, MgO is the most promising candidate [32,33,34,35] because of its non-toxicity. Unlike other basic metal oxides, the hydrolyzed form of MgO is also solid and has a negligible effect on rubber plasticity. Some attempts have already been made using nano magnesium oxide-based cure activator in the vulcanization of rubber [36,37,38,39], and the improvements in the properties are mainly due to nanoparticle reinforcement. Generally, nano cure activators are expensive due to complicated synthetic procedures, and also according to Ding et al. metal oxide nanoparticles are more toxic for aquatic organisms than conventional microparticles [40]. However, according to Kuschner et al. [41], micro magnesium oxide has very low environmental toxicity compared to zinc oxide. Tire industries always have a high demand for the simplest, easiest, and most economical way to reduce production costs with enhanced properties. Although magnesium oxide did not provide similar properties to zinc oxide, a significant improvement was possible by using binary accelerators instead of single-site curing accelerators [42]. In particular, binary accelerators, one of which contains zinc ions, can show synergistic effects on the vulcanization properties [42]. Hence, it is believed that MgO-only can, with some difficulty, replace the ZnO by using a single accelerator for the vulcanization of rubber. It was also concluded that MgO can undergo a reaction similar to ZnO with the vulcanizing accelerators. However, the lower cross-linking capacity of MgO could be due to a lack of active sulfurating complex, as was evident in the ZnO-based cure activator [42]. From different studies [11,12,13,14,15,16,31,32,38,42], it was revealed that partial or complete replacement of ZnO could be possible depending upon the purpose of application. For tire application, complete replacement of ZnO is quite impossible because the tires need the higher modulus and other advantages that cannot be achieved without ZnO. It was quite familiar that binary accelerators comprising thiuram and thiazole functional groups undergo mutual activity to produce higher vulcanization properties such as improved cross-link density, mechanical modulus, etc., than single accelerator systems in the presence of a zinc oxide-based cure activator [43,44,45,46,47]. Magnesium oxide as a cure activator also provided synergism on the vulcanization properties, but it was quite low due to the formation of a lower amount of cross-linking precursors [42]. However, in the presence of the zinc ion-containing accelerator, the synergistic activity was higher. Hence, it was believed that zinc compounds either formed in-situ or externally added may undergo interactions with MgO and can deliver the synergistic effect. Generally, lower zinc oxide can be useful in practice for a binary accelerators system where the accelerators undergo mutual interactions to produce higher cross-link density compared to single accelerator systems [42]. Thus, Guzmán et al. [33,34,35] studied the efficiency of single MgO and Zn/Mg oxide nanoparticles in reducing the amount of ZnO in the vulcanization of rubber. Interestingly, they found that mixed metal oxide nanoparticles improved the cross-link density and the rate of vulcanization compared to single activator systems [33,34,35]. However, instead of binary accelerators, they considered single accelerators for the vulcanization [33,34,35]. Recently, Alam et al. [42] found that, similar to ZnO, MgO could also promote the mutual interactions between thiuram- and thiazole-based accelerator systems to enhance the cross-link densities. Hence MgO as a co-cure activator with ZnO could be more effective in the binary accelerator systems than in single accelerator systems to reduce the amount of ZnO from the vulcanization. Moreover, MgO is quite cheap, nontoxic, and abundant. While most studies were done to reduce ZnO levels by nano cure activators or modified zinc compounds that could have additional toxicity, here we use conventional micro MgO, which is non-toxic. In this way, the rubber compounds are expected to have much less environmental toxicity. Since vulcanization is almost necessary for all practical applications with better properties, we use binary accelerators rather than single accelerators to obtain higher vulcanization properties followed by synergism with MgO.

In this article, we investigate MgO as a co-cure activator along with ZnO in the vulcanization of natural rubber. Low sulfur and a binary accelerators system are chosen to understand the mutual interactions between the accelerators and the activators. Moreover, a high accelerator to low sulfur ratio that is known as an efficient vulcanization system (EV), which can produce higher mono and disulfide cross-links with better thermal stability than other vulcanizing systems such as conventional vulcanization (CV) and semi-efficient vulcanization (SEV), is considered. Detailed curing, mechanical and thermal properties are investigated to establish the utility of MgO as a co-cure activator. Special attention is given to the fracture toughness of the rubber compounds, since this property is highly important for stretchable mechanical and electronic devices. Possible mechanisms of chemical interactions between the cure activators causing the synergistic activities on the properties are proposed and discussed in detail.

## 2. Materials and Methods

### 2.1. Materials

Zinc oxide, stearic acid, sulfur, and natural rubber (NR, RSS-3, density = 0.96 g/cm^3^) were supplied by the Thai Rubber Research Institute. Cure accelerators such as tetramethyl thiuram disulfide (TMTD) and dibenzothiazyl disulfide (MBTS) were purchased from Tokyo Chemical Industry Co., Ltd., Japan. Magnesium oxide light of fine powder was purchased from AppliChem PanReac, Thailand. X-ray studies of ZnO and MgO confirmed their excellent purities with hexagonal and cubic crystalline structures respectively. From the X-ray studies and using the Scherrer equation, it is confirmed that MgO bears a lower crystalline size compared to ZnO. The XRD plots of ZnO and MgO are given in Figure 1a,b.

### 2.2. Rubber Compounding

Natural rubber was first masticated in a laboratory size two-roll mill for 5 min to promote additives dispersion. After that, cure activator(s) and stearic acid were mixed for another 5 min. Finally, accelerators and sulfur were mixed for the last 5 min. After complete mixing, the compounded rubbers were cut as sheets. The friction ratio of the front and rear roller was maintained at a 1.2:1 ratio with roller speeds of 24 and 20 rpm, respectively. The nip gap between the two rollers was kept at 1 mm during mastication and the rest at 0.5 mm. The different mixing ingredients with the mixing formulations are provided in Table 1.

### 2.3. Measurements of Cure Characteristics

About 5 g of compounded rubber was placed in the cavity of a Moving Die Rheometer (MDR) to measure the curing characteristics at 140 °C. Moderate vulcanization temperature was used because some curing reactions were very fast. The MDR provided rheographs (torque vs. time curve) from which the different curing parameters were obtained. The different curing parameters such as lowest torque (M_L_), highest torque (M_H_), torque difference (M_H_ − M_L_ = ∆ torque), scorch safety time (t_2_), optimum curing time (t_90_), cure rate index (CRI = 100/ (t_90_ − t_2_)), and rate of vulcanization (R_v_ = (Mt_90_ − Mt_2_)/(t_90_ − t_2_)) were obtained from the rheographs.

### 2.4. Measurement of Cross-link Density of Rubber Vulcanizates

The cross-link densities of the vulcanized rubbers were determined by the swelling method and by applying the Flory-Rehner equation [48]. The rubber specimens were kept for swelling in toluene for 7 days to reach equilibrium swelling, and the cross-link densities were obtained as follows
V_c_= −{ln(1 − V_r_) + V_r_ + χV_r_^2^}/{V_s_d_r_(V_r_^1/3^ − V_r_/2)}
where V_c_ is the cross-link density, V_r_ is the volume fraction of rubber in the equilibrium-swollen specimen, V_s_ is the molar volume of solvent (toluene), d_r_ is the density of the rubber (0.96 g/cm^3^), and χ is the solvent-rubber interaction parameter.

The volume fraction of rubber in the equilibrium swelling stage was determined according to the formula
V_r_ = (W_r_/d_r_)/{(W_r_/d_r_) + (W_s_/d_s_)}
where W_r_ is the weight of dry rubber, W_s_ is the weight of solvent swelled, and d_s_ is the density of solvent. In this experiment χ = 0.3795, V_s_ = 106.2 cm^3^/mol, and d_s_= 0.87 g/cm^3^ were considered.

### 2.5. Mechanical Properties of Rubber Vulcanizates

The rubber compounds were cured corresponding to their t_90_ values in a hot press molding machine at 100 psi as sheets of 2 mm thickness. The cured rubber sheets were placed in a refrigerator to control the aging effects [45] due to the drastic changes in environmental temperature. Before measuring the tensile properties, the sheets were placed at ambient temperature for 24 h. Dumbbell-shaped test pieces were cut from the sheets according to the standard (ISO 37, type 2) to measure the tensile properties. The tensile tests were performed in a tensile testing machine (UTM, LLOYD LR 100 K, Lloyd Instruments, Hampshire, UK) using a 1 kN load cell and a cross-head speed of 500 mm/min. The gauge length of the dumbbell-shaped specimen was fixed at 25 mm. Tensile properties such as modulus at 100% elongation (M100), modulus at 300% elongation (M300), tensile strength (T.S), and elongation at break (E.B) were obtained from the stress-strain curves.

### 2.6. Scanning Electron Microscopic Analysis and Elemental Mapping

Scanning electron microscopic (SEM) analyses were performed on the tensile fractured surface of rubber samples by field emission scanning electron microscope (FE-SEM, S-4800, Hitachi, Tokyo, Japan). Elemental mapping to understand the dispersion of the curatives was achieved through the energy-dispersive X-ray spectroscopic technique. Before SEM analyses, the samples were pre-coated with platinum by a sputter coater.

### 2.7. Thermo Gravimetric Analysis

Thermal analyses were performed using a Thermo gravimetric analyzer (TGA, NETZSCH TG 209F3 TGA209F3A-0364-L) and heating the samples from 35 to 800 °C in a nitrogen environment at a heating rate of 10 °C/min in a crucible made of alumina.

## 3. Results and Discussion

### 3.1. Curing Characteristics

Cure curves (rheographs) of different rubber vulcanizates are provided in Figure 2a. From the rheographs, it is found that the nature of all the cure curves is more or less similar except for the vulcanizate with an MgO single activator system. After reaching the highest torque, MgO-only-based vulcanizate showed substantial reversion in the curing process. This suggests that MgO itself acts as a poor cure activator compared to ZnO alone. It is recognized that MgO may break the polysulfide linkages but is unable to reform as stable sulfur cross-links. However, the reversion can be completely removed by using binary curing activator systems. It is known that ZnO in the presence of a curing accelerator and sulfur produces an active sulfurating complex [49], whereas MgO does not produce such a type of active complex but it can decompose accelerators quickly and start the vulcanization early, as seen in Figure 2a. Moreover, MgO can decompose the poly-sulfidic bridges and suppress the total number of cross-links with increasing cure time. In the presence of ZnO and MgO combined cure activators, the vulcanization starts earlier, as does higher torque without reversion. Hence, it can be assumed that MgO as a co-cure activator may help to produce a zinc-accelerators complex more effectively than ZnO alone and that the degradation of sulfur cross-links caused by MgO can be suppressed completely. Since MgO can degrade higher-ranked sulfur bridges, a high accelerator-to-sulfur ratio, i.e., efficient vulcanization, should be the proper choice, rather than semi-efficient and conventional vulcanization systems [34,35,42] to get the benefits of MgO as a co-cure activator.

It seems that the M_L_ values are a little higher for MgO-based vulcanizates, which may be due to the faster rate of vulcanization. However, such little increases in the M_L_ values did not affect the flowing properties of compounded rubber during molding. The M_H_ values are increased with ZnO content in binary activator systems. The NR/3-MgO/2-ZnO shows an M_H_ value very near to that of NR/5-ZnO. This result suggests that, concerning M_H_ value, 60% ZnO can be replaced by MgO easily.

The torque differences for different vulcanizates are plotted in Figure 2b. From this figure, it can be found that the 3:2 ratio of MgO to ZnO in the binary activator system (NR/3-MgO/2-ZnO) provides similar Δ torque compared with 5 phr ZnO containing vulcanizate (NR/5-ZnO). A considerable increase in Δ torque from NR/5-MgO to NR/4-MgO/1-ZnO vulcanizate indicates that 80% ZnO can also be replaced by compromising a little lower value than that of the NR/5-ZnO vulcanizate. If we compare the Δ torque values of binary activator systems with single activator systems, prominent synergisms can be seen, which suggests that MgO may further activate the effect of ZnO on the cross-linking reactions.

Scorch safety is an important parameter for thick vulcanizate. The different scorch safety times are shown in Figure 2c. From this figure, it can be found that MgO-based vulcanizate bears a lower scorch safety value than ZnO-based vulcanizate. Actually, in the presence of MgO, the accelerators began to decompose and started cross-linking reactions much faster. On the other hand, ZnO reacts quite slowly with accelerators to form the active sulfurating complex, and then the cross-linking reaction starts at a faster rate. Hence, ZnO-only shows a higher scorch safety value than other compounds. Although MgO as a co-activator provides lower scorch safety values, these values can be improved by controlling the rate of vulcanization using a lower vulcanization temperature.

Optimum cure time is an important curing parameter, of which values below and above can result in lower vulcanizate properties. The different optimum cure times are shown in Figure 2d. From this figure, it can be seen that optimum cure times for MgO-based vulcanizates are lower compared to ZnO-only-based vulcanizates. Interestingly, the optimum cure time for 3 phr MgO as a co-cure activator is about 3 times lower than 5 phr ZnO containing vulcanizate, keeping similar Δ torque values. This implies that MgO as a co-cure activator with ZnO will be much more economical for rubber vulcanization than using ZnO-only as a single-site curing activator.

The cure rate index (CRI) values are plotted in Figure 2e. From this figure, it can be seen that MgO-based compounds have higher CRI values than ZnO-based single activator systems. It is also to be noted that a slight decrease in the CRI value can be found with increasing 2 phr to 5 phr ZnO content. From the equation, it can be seen that the CRI value depends on both scorch safety and optimum cure times. Since ZnO as a single-site curing activator poses a higher optimum cure time, the composites showed lower CRI values than MgO-based vulcanizates. The actual rate of vulcanization (R_v_) can be found in Figure 2f. Interestingly, although the MgO-only compound has a CRI value similar to that of the MgO/ZnO binary activators system, huge differences have been found in vulcanization rates among them. Moreover, the R_v_ value of the MgO-only-based activator system has a higher value than ZnO-only-based activator systems. These results suggest that MgO can improve the vulcanization rate, but the total cross-linking level is quite low, as is evident from the lower Δ torque. However, in the cases of MgO/ZnO binary curing activator systems, an increase in ZnO content increases R_v_ as well as Δ torque values. It is believed that in the presence of MgO, the accelerators decompose at faster rates, which increases the rate of vulcanization, and the decomposition of accelerators helps to produce a higher amount of zinc-accelerator complex, which efficiently vulcanizes the rubber.

The swelling index and cross-link density data are plotted in Figure 3a,b. From Figure 3a,b, it can be seen that the MgO/ZnO binary and ZnO-only cure activator systems provide a similar swelling index and cross-link density values. Moreover, MgO/ZnO binary activators provide much better curing efficiency than ZnO-only as a single-site curing activator in the vulcanization of rubber. MgO-only provides the highest swelling index and lowest cross-link density. The highest swelling index and low cross-link density suggest that MgO-only has poor efficiency in cross-linking.

The vital step in vulcanization is the formation of a metal complex combining the activator and the accelerator [50,51]. It is well known that zinc-dithiocarbamate is an ultrafast accelerator compared to corresponding thiuram disulfide [42]. It is well accepted that in the presence of ZnO, the thiuram-type accelerator forms a complex like zinc-dithiocarbamate [42,49,50,51]. This zinc-dithiocarbamate is then processed to an active sulfurating complex in the presence of sulfur and ultimately undergoes cross-linking. The detailed mechanistic aspects of sulfur vulcanization can be found in Figure 4 (steps 1–7) in the presence of ZnO and cure accelerators. Steps 1–5 (Figure 4) regard the formation of thiocarbamic acid, which undergoes an acid-base type reaction with ZnO. Since thiocarbamic acid is regarded as a weak acid and ZnO is a weak base, the formation of the zinc-dithiocarbamate type complex is very favorable. This dithiocarbamate undergoes an ionic exchange reaction with MBTS and again forms thiuram disulfide and produces the cross-links in steps 4 and 5 (Figure 4). It is believed that in the presence of ZnO and thiuram disulfide, a catalytic-type complex is formed. In the presence of sulfur, this active complex produces more and more cross-links between the rubber chains. It is believed that after cross-linking, the complexes return to dithiocarbamate, which itself has low cross-linking efficiency [44]. However, the presence of an oxidizing reagent, such as a secondary accelerator, can convert this dithiocarbamate to a more active in-situ thiuram disulfide [44]. It was observed that the rate of vulcanization was greater when thiuram disulfide was formed in-situ rather than being added externally [44]. This result suggests that, to improve the kinetics of vulcanization, in-situ conversion of dithiocarbamate to thiuram disulfide should be preferable.

The different possible steps in the presence of an MgO/ZnO binary curing activator are given in Figure 5. A similar type of zinc-dithiocarbamate, magnesium-dithiocarbamate can be formed in step 1 (Figure 5). This dithiocarbamate can undergo synergism with MBTS to produce higher cross-link density compared to single accelerator systems [42]. However, the number of cross-links achieved based on MgO as a single-site curing activator is much lower than based on ZnO as a single-site curing activator. It is believed that in the presence of MgO as a cure activator, a higher amount of magnesium-dithiocarbamate is formed; however, the resulting compound has no catalytic activity regarding the final cross-links. However, in the presence of ZnO, the magnesium-dithiocarbamate readily converts the zinc-accelerator complex of higher reactivity. Depending upon the amount of ZnO, the amount of conversion is also varied. The beauty of this conversion is that it produces a higher amount of zinc-accelerator complex at a higher rate than ZnO-only. In the presence of MgO, due to the higher basicity of MgO, the dithiocarbamate may produce at a faster rate but it has no catalytic activity with sulfur to form an active sulfurating complex, and hence it only enhances the rate of vulcanization. In this perspective, MgO can be assumed as a simple base that stabilizes the thiocarbamic acid and protects it from thermal decomposition (Figure 4, step 2). In the presence of the MgO/ZnO binary curing activator, the magnesium-dithiocarbamate finally converts to zinc-dithiocarbamate (Figure 5, step 3) which improves the cross-linking efficiency as well as the rate of vulcanization. From the concept of the acid-base theory of weak acids and weak bases, the reaction in step 3 (Figure 5) is highly feasible.

### 3.2. Tensile Mechanical Properties

The different tensile mechanical properties are plotted in Figure 6a–f. The most representative of the average stress-strain curves is provided in Figure 6a for different vulcanizates. From this figure, we can roughly say that MgO-only as a cure activator provides a lower overall modulus and higher elongation at break values compared to other vulcanizates. The specific modulus such as M100 and M300 in Figure 6b,c have similar trends for all the vulcanizates. It can be noted that the MgO/ZnO binary activator at a 3:2 ratio of MgO to ZnO provides the best modulus values (0.58 MPa in M100 and 1.36 MPa in M300) among the compounds. Regarding tensile strength in Figure 6d, the MgO/ZnO binary activator at a 4:1 ratio of MgO to ZnO provides the highest tensile strength (24.8 MPa). Interestingly, it can be noted that MgO-only as a cure activator can achieve a similar tensile strength value compared to ZnO as a single-site curing activator system. The elongation at break values for different compounds is shown in Figure 6e. From this figure, it can be seen that elongation at break values decreased with the increase of ZnO content in the binary activator systems. Fracture toughness is an important mechanical property that is necessarily useful for stretchable electronic applications [52,53]. Regarding the toughness value, MgO-only and binary activator systems provide better toughness values compared to ZnO-only as a cure activator. The highest toughness value (132 MJ/m^3^) was obtained for the MgO/ZnO binary activator system at a 4:1 ratio of MgO to ZnO content. The better toughness values of the binary activator systems might be due to a better modulus and elongation at break values. From the above discussion, it can be concluded that a complete ZnO-free vulcanizate can be possible where high toughness is necessary, sacrificing the modulus values.

Boonkerd et al. described the relationship between tensile properties with the number of sulfur cross-links and the sulfur ranks per cross-link [54]. A greater number of cross-links with a lower sulfur rank indicate low elongation and a higher modulus. On the other hand, at similar cross-link density, a higher sulfur rank (i.e., polysulfide cross-links) enhanced the tensile strength and elongation at break values. It is believed that elongation at break values is highly dependent on the dispersion of the curatives as well as cross-link density and sulfur ranks among the cross-links. From the cross-link density measurements and tensile properties, it can be predicted that higher MgO content provides higher sulfur-ranked cross-links and excellent dispersion of the curatives. Good dispersion of curatives not only improves the cross-link density but also enhances the physical bonding between the remaining unreacted or vulcanization byproducts and the rubber molecules that have some contribution to the enhanced mechanical properties. Since the toughness is related to the number of total linked bonds and their strengths, it can be assumed that the highest number of linked bonds exist in the MgO-only-activated system reacting with the most sulfur elements in the system. On the other hand, binary activator systems provide a higher number of linked bonds as well as a higher number of stronger C-S bonds that assure excellent toughness and modulus values compared to single-cure activator systems. According to Borros et al. [33], when both zinc and magnesium existed in the vulcanization, a higher amount of disulfide linkages were obtained than mono and polysulfide linkages [33]. For the existence of disulfide linkages in higher amounts in the vulcanized compound [54], excellent dispersion of curatives (Figure 7), a smaller particle size of MgO, a better modulus and tensile strength were obtained in MgO/ZnO binary systems. Thus, magnesium oxide plays an important role in improving the tensile properties, especially the tensile strength and fracture toughness of the rubber vulcanizates.

### 3.3. SEM and EDS-Mapping

The distribution of curatives in the rubber matrix is an important factor that ensures homogeneity in the cross-link density throughout the rubber compounds. It is believed that more homogeneity in the cross-links throughout the rubber matrix provides better tensile properties. Figure 7a–c represents the SEM images of single and binary curing activator systems. From Figure 7a, it is evident that some curatives remain separated from the matrix because of their incomplete chemical interactions with the rubber matrix in the presence of MgO as a cure activator. Similarly, in the presence of a ZnO-only as a cure activator more curatives remain unreacted and are separated from the rubber matrix (Figure 7b). On the other hand, the binary curing activator ensures excellent distribution and reactivity of all curatives, and only a few remain unreacted (Figure 7c). These results are highly correlated with the improved toughness in the MgO/ZnO binary curing activator systems. To confirm the homogeneous distribution of curatives in the MgO/ZnO binary curing activator system, EDS mapping for different elements is performed and is presented in Figure 8. EDS-mapping for different elements indicates the homogeneous distribution of curatives for the 3:2 ratio of MgO to ZnO-based binary curing activator system that ensures better tensile mechanical properties than ZnO-only as a single-site curing activator in the vulcanization of rubber.

### 3.4. Thermo Gravimetric Properties of Rubber Vulcanizates

The thermal properties of conventional 5 phr ZnO-based vulcanizate were compared with 5 phr MgO-based and binary curing activators containing 3 phr MgO and 2 phr ZnO-based vulcanizates. Literature showed that the main products of thermal degradation of NR are isoprene, dipentene, and *p*-menthene [55]. The weight losses from ~150 to ~327 °C in Figure 9a,b, are mainly due to decomposition of the un-reacted crosslink precursors, sulfur cross-links, stearic acid and accelerator, and partial breakage of the rubber backbone [56]. The rapid degradation region from ~350 to ~450 °C in the thermal gravimetric analysis (TGA) curves in Figure 9a represents the main chain breakdown of rubber polymer. This behavior is clearer on the derivative thermo-gravimetric analysis (DTA) curves in Figure 9b. From these figures, it is clear that there are no significant differences in the thermal stabilities of the compounds cured by single MgO and MgO/ZnO binary curing activators compared to 5 phr ZnO as a single-site curing activator. The comparable thermal stabilities may be due to better homogeneity in the cross-links and reduced thermal motion of rubber chains in the presence of MgO [38,57] compared to ZnO-only.

## 4. Conclusions

In this research non-toxic MgO was tested as a cure activator for the sulfur vulcanization of rubber. We aimed to reduce the amount of environmentally hazardous ZnO cure activator without compromising curing and mechanical properties in the vulcanization of rubber. The curing, mechanical, and thermal properties were investigated to find out the suitability of the MgO by itself or in combination with ZnO in the vulcanization of rubber. Results revealed that considering some advancements in properties like curing time, elongation at break, and fracture toughness, MgO can compete with the conventionally used ZnO as a cure activator. However, to achieve the present industrial level of cross-link density, mechanical modulus, and tensile strength, the binary combination of MgO and ZnO should be used as curing activators. Considerable reductions (60–80%) of ZnO from the conventional amount could be possible by utilizing an MgO/ZnO-based binary curing activator system with improved efficiencies in the curing kinetics and mechanical properties with negligible reduction in the thermal stability. For example, 60% MgO in MgO/ZnO binary activators system provides 0.58 MPa of M100, 23.7 MPa of tensile strength, 1455% of elongation at break, 126 MJ/m^3^ of fracture toughness, and 0.85 Nm/min of vulcanization rate values, which are 7.5%, 20%, 15%, 68%, and 184%, respectively, higher than the 100% ZnO-based activator system. Investigations on the curing properties suggest that certain chemical interactions might have happened between the two cure activators following the paths of accelerators-activators complex formation reactions. Instead of a single-site curing activator, the proposed binary curing activators could be very useful in the rubber industries for considerable improvements in the vulcanization kinetics and tensile properties.

## Figures and Tables

**Figure 1 polymers-14-05289-f001:**
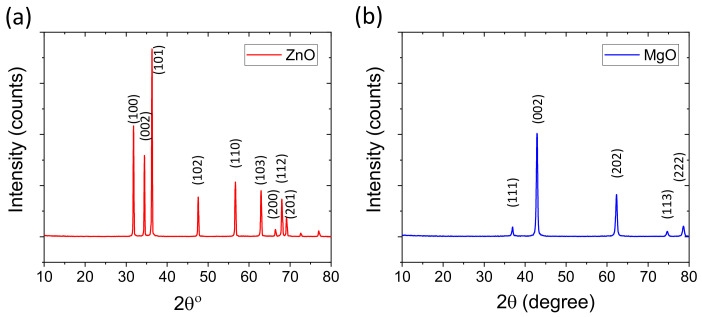
XRD plots of used cure activators; (**a**) ZnO and (**b**) MgO.

**Figure 2 polymers-14-05289-f002:**
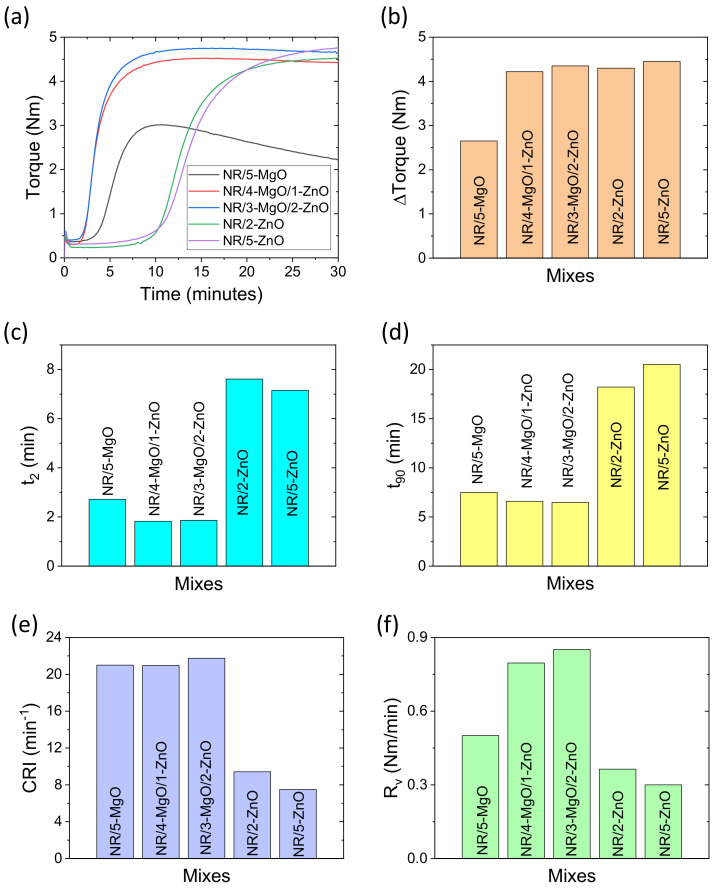
Curing characteristics of rubber vulcanizates; (**a**) rheographs, (**b**) Δ torque, (**c**) t_2_, (**d**) t_90_, (**e**) CRI, and (**f**) R_v_.

**Figure 3 polymers-14-05289-f003:**
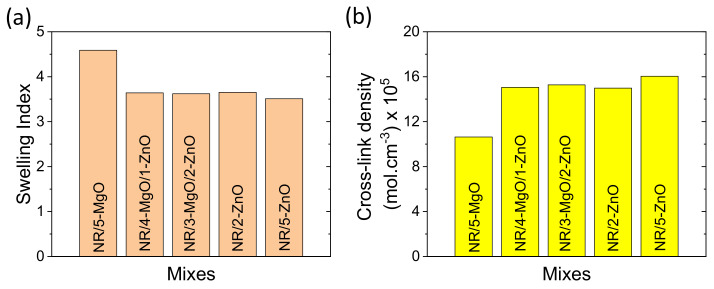
(**a**) Swelling index and (**b**) cross-link density of rubber vulcanizates.

**Figure 4 polymers-14-05289-f004:**
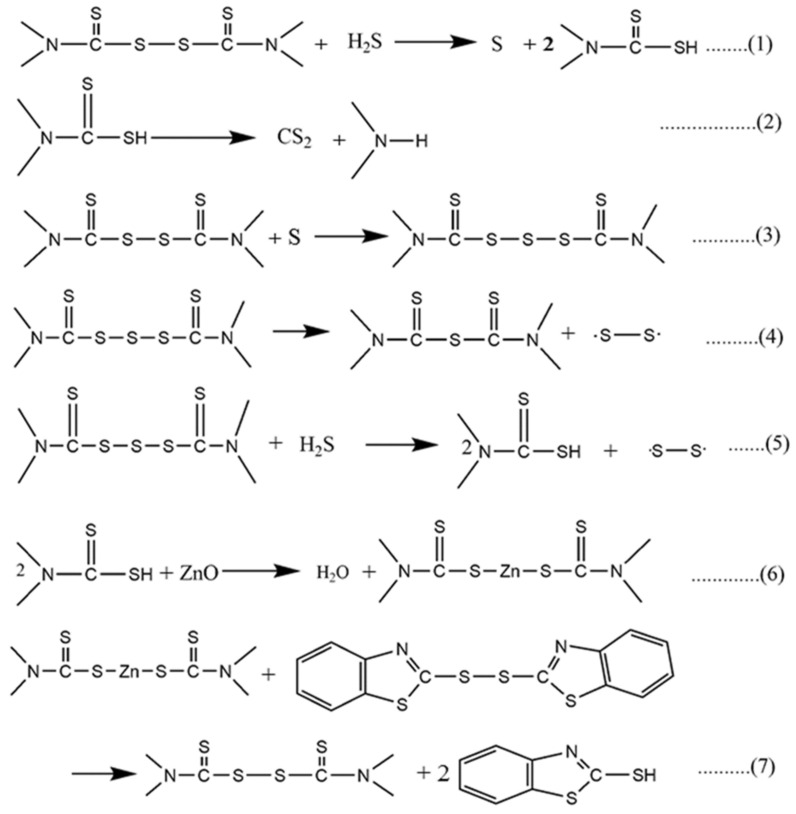
Mutual interaction of thiuram- and thiazole-based accelerators in presence of ZnO cure activator.

**Figure 5 polymers-14-05289-f005:**
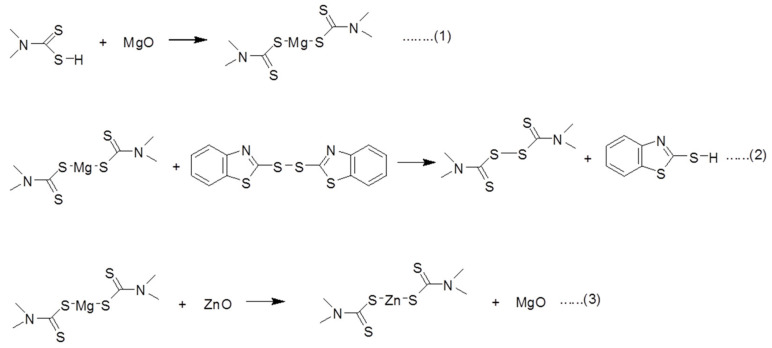
Possible chemical interactions between MgO and ZnO in the binary curing activators systems.

**Figure 6 polymers-14-05289-f006:**
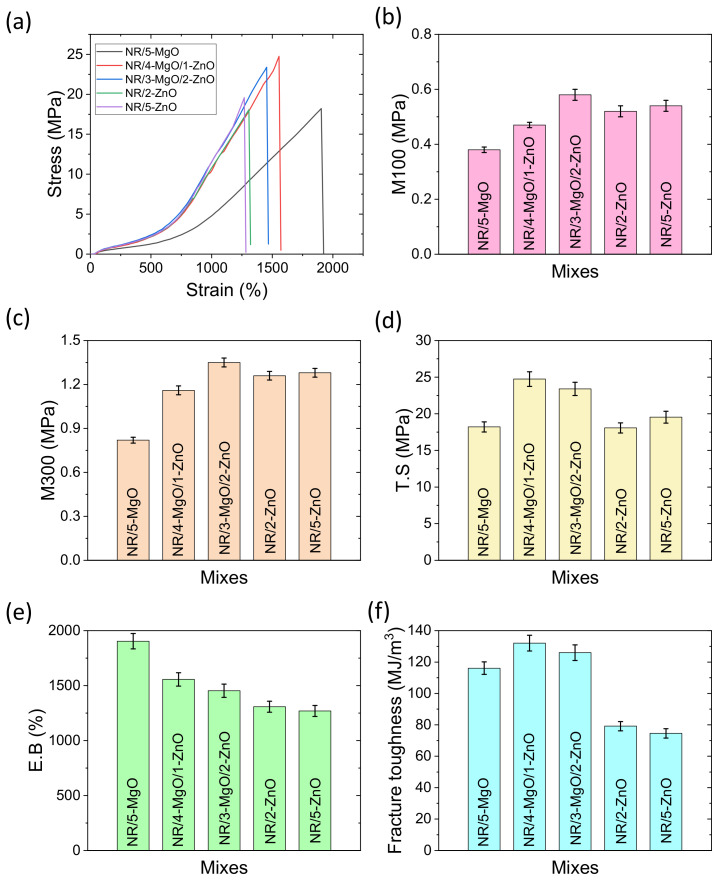
Tensile mechanical properties of vulcanized rubber; (**a**) stress-strain, (**b**) M100, (**c**) M300, (**d**) T.S, (**e**) E.B, and (**f**) fracture toughness.

**Figure 7 polymers-14-05289-f007:**
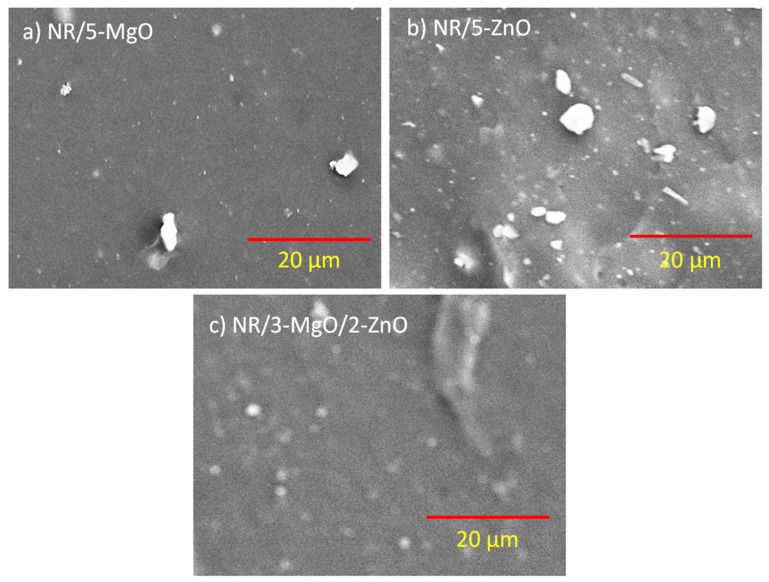
SEM micrographs of vulcanized rubber; (**a**) NR/5-MgO, (**b**) NR/5-ZnO, and (**c**) NR/3-MgO/2-ZnO.

**Figure 8 polymers-14-05289-f008:**
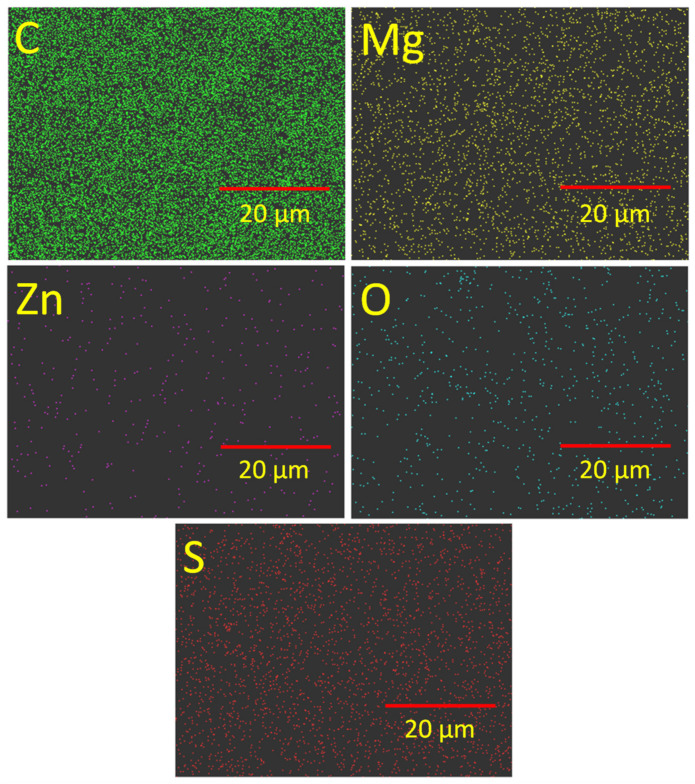
Particle distribution through EDS-mapping in NR/3-MgO/2-ZnO compound.

**Figure 9 polymers-14-05289-f009:**
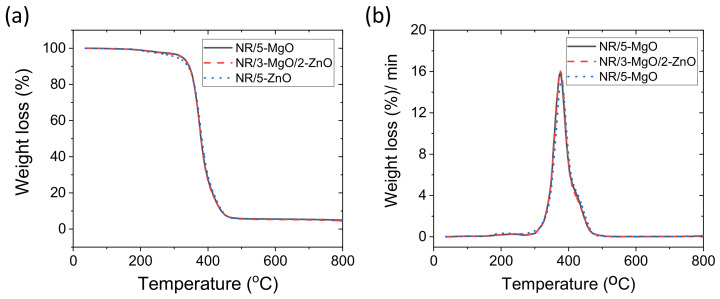
(**a**) TGA and (**b**) DTA results of vulcanized rubber.

**Table 1 polymers-14-05289-t001:** Mixing composition of different ingredients in phr (per hundred gram of rubber).

Formulation	Mixing Composition				
	NR/5-MgO	NR/4-MgO/1-ZnO	NR/3-MgO/2-ZnO	NR/2-ZnO	NR/5-ZnO
NR	100	100	100	100	100
MgO	5	4	3	0	0
ZnO		1	2	2	5
Stearic Acid	2	2	2	2	2
TMTD	0.72	0.72	0.72	0.72	0.72
MBTS	2	2	2	2	2
Sulfur	0.5	0.5	0.5	0.5	0.5

## Data Availability

The data presented in this study are available on request from the corresponding author.

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
