# Peer review of "Advances in Rubber Compounds Using ZnO and MgO as Co-Cure Activators"

_polymers, 2022, doi:10.3390/polym14235289_

Round 1
Reviewer 1 Report
see review attached.

Author Response
Reviewer 1
General comments
This manuscript presents the development of natural rubber (NR) recipes vulcanized with a regular efficient (EV) sulfur/accelerant system, but partially replacing ZnO (one activator) by another metal oxide, such as MgO. The importance of this strategy lies in the growing interest of Rubber Science and Technology in achieving circularity in elastomeric materials. It is known that the production of many of the additives in a typical rubber recipe has a considerable environmental impact, in addition to the impact generated at the end of the lifetime of the final product. One of the most problematic additives is ZnO, so in recent years strategies have been sought to replace it totally or partially with other more sustainable additives or to give them an appropriate role within the rubber recipe (e.g., as a vulcanization agent in dynamic and sustainable ionic systems). The manuscript is interesting and present important advances, but requires major revisions before being accepted for publication in Polymers.
Authors’ reply: Thank you very much for your careful reading and providing your most valuable comments and suggestions. Please follow the revised manuscript where we made corrections accordingly highlighting with red colors.
Specific comments
- I suggest a thorough revision of the grammar and wording of the manuscript.
Authors’ reply: We have modified our manuscript accordingly. Thank you for you valuable suggestion.
- Abstract
- I would recommend including in the Abstract some specific values (and/or conclusions) that were obtained for the properties described in that section (modulus, tensile strength, fracture toughness, elongation at break, thermal stability...). For example: “With the addition of the MgO, an improvement of X % in tensile strength was obtained, going from X MPa to X MPa, respect to pure ZnO compounds”. This would be useful to put the reader in context about the improvements achieved and the information that can be found in the main text.
Authors’ reply: Thank you very much for your careful reading and help us to improve our manuscript. We have modified our manuscript accordingly in the proper section in the revised manuscript.
- Introduction
- Based on the literature provided, the authors seem to have overlooked some more recent research and key references published in high-impact editorials about the role of ZnO in the vulcanization process of different rubbers that ties perfectly with the research conducted. Here are some interesting references that can be incorporated in the Introduction section. I encourage authors to expand this list with a search of their own and provide a more recent point of view (1-5 years) of the research in this area.
- a) Polymer Bulletin, 2022, 79, 8535-8549, DOI: 10.1007/s00289-021-03921-5
- b) Composites Communications, 2021, 25, 100755, DOI: 10.1016/j.coco.2021.100755,
- c) Polymers, 2021, 13, 19, 3234, DOI: 10.3390/polym13193234,
- d) ACS Applied Materials and Interfaces, 2020, 12, 42, 48007-48015, DOI: 10.1021/acsami.0c15114,
- e) Catalysts, 2019, 9, 8, 664, DOI: 10.3390/catal9080664,
Some questions can be answered to improve the introduction: What other role does ZnO play in rubber vulcanization? What can be extracted from such recent research? How does this study differ from others in your area? What is the importance of rubber vulcanization?
- The authors state that one of the main limitations of ZnO is its impact on the environment, especially in aquatic environments. However, no information related to its alternative is presented. How is MgO in this respect? Including some comparative data on the sustainability of both oxides may help to better illustrate the problem and MgO potential.
Authors’ reply: Thank you very much for your careful reading and providing your important comments and valuable suggestions. The suggested references were very interesting for further developing the quality of our paper. MgO is much less environment pollutant than ZnO according to reference 41. Please follow the revised introduction part to get answers of your suggestions and queries.
- Experimental
- Please specify the grade of the NR used.
- In the recipes used, the authors have selected an EV system (high accelerant content in relation to sulfur). This will tend to the formation of a predominant network of monosulfide and disulfide bonds. What is the motivation for selecting this system and not a conventional (CV) or semiefficient (SEV) system?
- 140 ºC seems a low vulcanization temperature for sulfur-based systems. What was the motivation for selecting this value?
- Why were the vulcanized rubber samples placed in a refrigerator? This is more common in non-vulcanized samples (to avoid vulcanization), but in vulcanized systems it is not common.
- Some details can still be specified to improve the methods section, for example:
o What is the value of the rubber density (dr) used for the calculation of the crosslink density? How was dr determined?
o What was the pressure used for vulcanization in the hot press?
o Are the stress-strain curves presented averages or the most representative of the average?
o Did the SEM observation require pre-coating? Specify.
Authors’ reply: We have modified our manuscript accordingly. Please find it in the experimental section. We are highly appreciate your valuable comments and grateful to you for your careful reading.
- Results and Discussion
- Please, add the thermogravimetric curve of NR/5-MgO to Figure 9 and comment. We see that the partial substitution of ZnO does not affect the stability of the system with pure ZnO, but how is it with respect to pure MgO?
- My only concern stems from the crosslink density results and their relationship to mechanical properties. One would expect that the higher the crosslink density, the better the mechanical performance, at least higher moduli. However, the crosslink density values obtained for the ZnO/MgO and pure ZnO are practically similar, but the moduli and tensile strength show greater differences. Why could this be? Could the activator be influencing the type of bond produced during vulcanization? I am aware that the authors mention some previous research by Borros et al (REF 26) and Boonkerd et al (REF 46) to explain this, butit is still not entirely clear. Including some illustrative schematic of the types of networks as an additional figure might be helpful. What techniques could be used to deepen this area? Adding a sentence about this and the need for specific studies with other techniques, such as nuclear magnetic resonance (NMR), could be useful too.
Authors’ reply: We have added TGA of NR/5-MgO and got similar thermal stability. Hence we believe that, here, the thermal stability mostly controlled by partially magnesium oxide and partially by mono-sulfidic linkages as we use EV vulcanization system.
We believe that although the cross-link density of ZnO/MgO hybrid is slightly low but the MgO oxide poses better dispersion that may enhance the physical interaction between the unreacted MgO particles and the rubber molecules and enhance the modulus than 100% ZnO. Also most of the sulfur used might be reacted with rubber chains and produced longer connective chains in presence of MgO due to high reactivity. Although we assumed that the higher disulfidic cross-links were formed MgO-based activators according to the references but we wish to check the cross-linking structures in our systems. Unfortunately we don’t have NMR facility and indeed we believe that providing the illustrations of predicted cross-linking structures is not right at this moment. Hope you will agree with us and we will keep your valuable suggestion for our future publication. If you have any further suggestion please let us know.
- Conclusions
- The conclusions reflect very well the work described. I would simply suggest including some specific values to quantify the different achievements in mechanical properties, curing kinetics, etc.
Authors’ reply: We have revised the conclusion section according to your valuable suggestion. Again than you very much for your careful reading and providing valuable comments and fruitful suggestions.

Reviewer 2 Report
This manuscript addresses an old but still unsolved problem of vulcanizates (especially in car tires), namely the contamination of tire wear and scrap tires with zinc oxide, which is still a necessary activator in virtually all sulfur-cured rubber products.
This issue has recently been highlighted by new research on microplastics in the environment, which found that particles from tire wear are the largest source of so-called microplastic particles, particularly in lakes, rivers and the ocean. This is striking evidence of the full extent of ZnO's toxicity to aquatic life.
The environmental significance of this study cannot be overstated.
In the introduction, the authors describe in very comprehensive terms the state of the art in avoiding or reducing ZnO in elastomers.
The partial replacement of ZnO by MgO is described clearly and well understandably. All statements are experimentally well supported.
A few orthographic errors should be corrected before publication.
Otherwise, I can recommend the publication of the manuscript in its present form.
Author Response
Reviewer 2
This manuscript addresses an old but still unsolved problem of vulcanizates (especially in car tires), namely the contamination of tire wear and scrap tires with zinc oxide, which is still a necessary activator in virtually all sulfur-cured rubber products.
This issue has recently been highlighted by new research on microplastics in the environment, which found that particles from tire wear are the largest source of so-called microplastic particles, particularly in lakes, rivers and the ocean. This is striking evidence of the full extent of ZnO's toxicity to aquatic life.
The environmental significance of this study cannot be overstated.
In the introduction, the authors describe in very comprehensive terms the state of the art in avoiding or reducing ZnO in elastomers.
The partial replacement of ZnO by MgO is described clearly and well understandably. All statements are experimentally well supported.
A few orthographic errors should be corrected before publication.
Otherwise, I can recommend the publication of the manuscript in its present form.
Authors’ reply: Thank you very much for your careful reading of our manuscript and providing valuable suggestion. We are happy to include one reference [ref. 10] regarding the ZnO pollution from tire wear. This is very important suggestion we got from you and we will try with our best to work on tire wear to protect the environments in future. Please find your suggestion in the revised manuscript.

Round 2
Reviewer 1 Report
The authors have addressed all the issues raised by the reviewers. Therefore I am happy to accept the manuscript in its current form for publication.